# Sustainable Adoption of E-Learning from the TAM Perspective

Saman Sattar Saleh * , Muesser Nat and Musbah Aqel

Management Information Systems Department, Cyprus International University, Via Mersin 10,
Haspolat-Lefkosa 99258, Turkey; mnat@ciu.edu.tr (M.N.); maqel@ciu.edu.tr (M.A.)
* Correspondence: samsofteng2017@gmail.com

**Abstract:** This study investigates the imperative to adopt e-learning and how it influences educational process sustainability. For developing countries, adopting e-learning has always been a challenge because of the lack of mechanisms due to the resistance of teachers and students, low level of information and communication technology (ICT) literacy, and lack of ICT infrastructure. A quantitative research methodology was used by applying a hypothesized extended model of the technology acceptance model (TAM) for the adoption of e-learning. The factors were investigated by analyzing the intentions of 367 students and teachers. Data were collected through an online questionnaire. This study aims to identify the factors that influence students' and teachers' attitudes toward the adoption of sustainable e-learning and illustrate the moderating role of the mentality acceptance factor in the relationship between system trust and interaction, on the one hand, and PU and PEOU on the other. Findings of the study indicated that all TAM constructs significantly impact the BI of e-learning use. Additionally, the results showed that mentality acceptance substantially moderates the relationship between system trust and interaction, on the one hand, and PU and PEOU on the other. These findings suggest that educational institutions should focus on factors influencing teachers' and students' attitudes toward adopting and using e-learning services. Lack of internet connection, ICT skills, and technology capabilities are the main issues, and the main TAM constructs of all factors.

**Keywords:** e-learning; TAM; adoption and sustainability





## 1. Introduction

E-learning has made significant strides in improving educational quality worldwide [1]. Our economies and our entire administrative and educational institutions are being reshaped due to the rapid expansion of digital technologies. Artificial Intelligence (AI), robotics, business analytics, and virtual reality contribute to successful digital learning by enabling end-to-end digital connectivity [2]. According to the authors of [3], perceived competitiveness and perceived risk significantly impact the intention to use Industry 4.0 processes, which are primarily based on digital technology. The authors of [4] investigate how several factors interact to assist businesses in becoming more strategically resilient, with digitization and agility serving as enablers in the process.

To describe the use of technology in educational situations to improve academic performance, the term "e-learning" is used [5]. Education is seen as a critical component of gaining a competitive advantage in today's technological era because it encompasses teaching and learning. Knowledge-intensive individuals and businesses are gaining a competitive advantage by spending more time and resources collecting and retaining information. ICT improves educational standards by using technology to help students learn more effectively [6]. In schools, e-learning has numerous advantages. An essential benefit of adopting online learning is that students have the flexibility to access course materials at any time and from any location. Student participation in active learning is encouraged as well. The utilization of video and audio broadcasting technologies, personal

computers, the World Wide Web, movies, and PowerPoint slides creates an engaging learning environment for students participating in distance education [7].

People's propensity to adopt new technologies is influenced by various factors, including their familiarity with technology acceptance models such as the Technology Acceptance Model (TAM) [8]. Understanding the difficulties students confront and their incentives for using e-learning, to support long-term adoption, is critical. An instructional environment, an e-learning experience, an ineffective learning experience, and limited contact between students and teachers are all challenges that need to be addressed in learning modalities [9]. Research shows, in earlier studies, that e-learning programs have difficulty succeeding [10].

"Sustainably education" is a method of teaching that incorporates sustainability principles, while also increasing students' understanding of environmental issues [11]. Both the general public and specific professions need to be educated about sustainability. When it comes to making decisions and managing human resources, there has been an increasing focus on sustainability education and related literature [12]. Several issues have emerged, including those that have made it challenging to sustain e-learning efforts and long-term implementation.

Education in Northern Iraq is being improved by implementing a modern and sustainable e-learning system incorporating cutting-edge information and communication technologies. The Northern Iraq E-Learning Leaders' Project was started in 2020, following the introduction of the COVID-19 crisis. This study revealed numerous barriers to the widespread adoption of e-learning and its long-term viability. Students, instructors, and school authorities alike expressed their dissatisfaction with the educational system in Northern Iraq due to a lack of familiarity with and understanding of this approach. To obtain the most out of e-learning, both students and instructors must have faith in it and the environment in which it is delivered. To improve people's comprehension and understanding of e-learning sustainability, the findings of this study will be helpful. This research aims to reveal the elements that influence e-learning in Northern Iraq and how they are perceived. This study employs a well-known theoretical concept, the TAM, for adoption. E-learning adoption and long-term viability in Northern Iraq is the main focus of this research.

## 2. Research Context

This study focuses on an area in Northern Iraq that is still developing in the interest of creating a better education model including e-learning. In addition to looking at how e-learning is being used in the institutions, the study also examines the significant obstacles to its general use. This study takes place in the newly formed Iraqi federal territory of Northern Iraq, located in Western Asia.

Students and instructors in the Northern Iraqi Governorate were surveyed using a quantitative method to examine their utilization of e-learning. Negative or critical frameworks are necessary for quantitative approaches [13]. According to this theory, people can measure, analyze, foresee, and generalize the concepts that pertain to their world, which is stable and reasonably coherent [14]. A well-structured quantitative approach has well-defined procedures that are identifiable [14]. To measure reality and facts, quantitative methods rely on these assumptions. Quantitative methods utilize techniques for measuring and testing variables, and aid in the discovery of how one variable affects others and how people feel and think about a specific notion or phenomenon. They also show how many people have proposed a particular idea. Quantitative approaches rely on numerical data studied using statistical processes [15]. Most students and instructors are not aware of the most significant challenges and barriers to the acceptance of sustainable e-learning services; thus, this is a critical step for them to take. Accordingly, the main research question of this study is: What are the most essential (partial) elements affecting the adoption of long-term e-learning applications by the Northern Iraqi government?

## 3. Literature Review

### 3.1. Technology Acceptance Model, E-Learning Adoption, and Hypotheses

The TAM was used in this study as an information-theoretic model. This method predicts the reception of new technologies by using specific user groups. System integration becomes increasingly challenging as the need for technology increases and computerization continues. Therefore, the user acceptance of individual technologies has become a research subject. We decided to include a technical component in the Theory of Planned Behavior (TPB) to suggest a strategy for gaining approval for technological innovations. This study aims to identify and eliminate the reasons for these technical issues [8]. Everyone has a role in advancing sustainability education because every discipline is involved. E-learning's success depends on three factors: its need, the talents it requires, and its requirements. Achieving long-term digital learning-development sustainability gives the educational institution the ability to overcome any challenges. Dissemination of e-learning applications is a priority for various institutions, regardless of their cultural background or physical infrastructure requirements. The first step is to identify the obstacles that prevent the system from working [16]. This is despite researchers [17] discovering that e-learning often involves transitions, comprehending computers, and internal organization infrastructure. As a result, we must start with a metric to establish readiness for e-learning. If implemented effectively, incorporating technology into classroom instruction may be a powerful teaching and learning tool [18].

According to the literature, some studies have successfully broadened the application of the TAM in e-learning technologies [19–21]. Traditional pedagogy is practiced despite the absence of financial resources and skilled staff in most underdeveloped countries [22]. According to TAM, attitudes regarding the adoption of new technologies are directly influenced by how practical and straightforward new technologies are seen to be. In addition, the attitudes of users, which can be characterized by their level of interest in specific systems, is a significant consideration. Therefore, the future use of these systems depends on the users' behaviors [23]. People's behavior is also affected by how easy and helpful a product is for them to use [24].

People's behavior is influenced by their perception of a product's usability and usefulness [24]. The designs of e-learning studies highlight the connection between system design and utilization, which is of particular interest. In the absence of a direct correlation, the link between PU and PEOU affects behavioral intention. Attitude toward adopting technology (ATT) is sometimes considered a direct predictor of perceived ease of use (PEOU) [25]. People's willingness to employ technology is influenced by technology's perceived usefulness (PU) [26]. Whether raising PEOU enhances one's perception of its usefulness has also been investigated [27]. PU has been proposed as a factor that directly influences ATTs, increasing the likelihood that people will engage with technology more frequently [20]. As a result, a person's attitude is characterized by how they react to new technology. Previous research has shown that a positive attitude towards new technology is necessary for successful implementation [28]. As a result, it has been proposed that ATT is a crucial determinant of different use behaviors [5]. PEOU, PU, attitude, and desire to use an e-learning service can be described by the following six hypotheses:

**Hypotheses 1 (H1).** *Perceived ease of use (PEOU) significantly affects the perceived usefulness (PU) of e-learning.*

**Hypotheses 2 (H2).** *Perceived ease of use of e-learning will lead to an increasingly positive attitude towards e-learning.*

**Hypotheses 3 (H3).** *Perceived ease of use of e-learning significantly affects behavioral intention to use e-learning.*

**Hypotheses 4 (H4).** *Perceived usefulness of e-learning has a positive effect on attitudes toward e-learning.*

**Hypotheses 5 (H5).** *Perceived usefulness of e-learning significantly affects behavioral intention to use e-learning.*

**Hypotheses 6 (H6).** *Behavioral intention to use e-learning will increase with a positive attitude.*

*3.2. Trust in the System, the Perceived Ease of Use, and the Perceived Usefulness of E-Learning Services*

The adoption of e-learning services was studied using a modified TAM. Trust is emerging as a potential driver of information technology (IT) adoption. People in today's uncertain and ever-changing Internet environment look to the trust theory for guidance. As defined by numerous academic criteria, trust enhances the usability of online learning services and public perceptions. According to [29], e-learning and supporting technology are essential [30]. A person's willingness to use and adapt to new technology is strongly influenced by their level of trust in the system and when people trust a system, they use technical services that are more convenient and useful [31,32]. Trust is thought to influence perceived usefulness, and ease of use is believed to be affected by trust as well.

**Hypotheses 7 (H7).** *There is a direct and positive link between trust in the system and the perceived usefulness of e-learning services.*

**Hypotheses 8 (H8).** *There is a direct and positive link between trust in the system and the perceived ease of use of e-learning services.*

This study uses the TAM, which includes interaction as a component. Interaction is a common topic in e-learning literature. Most studies have found that e-learning outcomes can only be achieved through direct interactions with students. According to this, interaction cannot be replaced with a structure. Numerous empirical studies have demonstrated that students' enjoyment and perceptions of educational quality are heavily influenced by contact [33]. As stated by Priyadarshini and Bhaumik [34], students' acceptance of e-learning is affected by the authenticity of the information, services, systems, and teachers. Therefore, infrastructure can boost the perceived utility and usability of e-learning to understand better how e-learning services are perceived as valuable and easy to use. Thus, this study hypothesized that:

**Hypotheses 9 (H9).** *There is a direct and positive link between interaction and the perceived usefulness of e-learning services.*

**Hypotheses 10 (H10).** *There is a direct and positive link between interaction and perceived ease of use of e-learning services.*

According to user impressions, individual mentality acceptance is strongly linked to using technology in everyday tasks. According to the findings of this study, individuals conduct their utility assessments based on how comfortable they are using technology in their daily routine. As e-commerce is simple, some people may develop skepticism. A technical use case that enhances performance can increase a technology's perceived usefulness [35]. It is important to remember that an individual's level of motivation influences e-learning. When a person initially embraces it, there is also a lot of social support available to them [36]. According to the data supplied, people who lack confidence in their talents are more likely to use e-learning. Students' abilities to direct and customize their educational experiences may boost their senses of well-being and motivation to continue learning [37].

Al-adwan and Smedley [38] found that the acceptance of e-learning can be attributed to various factors, such as students' growing appreciation for and knowledge of ICT's potential to enhance education and students' development of appropriate mindsets to take advantage of ICT's educational benefits. Studies have acknowledged that new learning frameworks are accepted by those that reveal positive attitudes towards e-discovery. New skills and a better academic future are made possible through online learning. E-learning is popular among students with specialized abilities [39]. There has been an increase

in professors' and students' requirements to modify their mindsets in light of distance learning, online courses, and other e-learning features [40].

Furthermore, ref. [41] noted that mindsets ultimately shape the strategies governments pursue to introduce technology-enabled educational applications regarding e-learning and the information society in these countries. However, with appropriate financial and logistical assistance, policies can change people's perspectives and attitudes regarding the need for technological innovation. Perceived simplicity of use, social norms, mind-stimulating playfulness, and product characteristics positively impact perceived usefulness [42]. Other factors contributing to ease of use are societal standards and product attributes. User behavioral intention is positively influenced by perceived effectiveness, ease of use, social norms, and network externalities.

Researchers have also examined how attitude acceptance influences system trust, PEOU, and PU. As a result, both interaction and PEOU and interaction and PU had moderating effects. Students and teachers used e-learning when they felt confident. Nevertheless, cyber-ethical issues, such as hacker exploitation, sabotage, vandalism of another person's physical, electronic, or intellectual property, and exploitation of the information will impact people's trust in online education. Our goal was to determine how user expectations and system confidence interact to determine whether attitude acceptance is significant. Students and teachers with the same level of trust in the system and the internet are more likely to profit from e-learning services. However, e-learning services may not be used if ethical issues exist. For the long-term, sustainable adoption of e-learning, acceptance of mentality may have a favorable moderating influence on the link between trust, on the one hand, and PEOU and PU, on the other.

This study presents the following hypotheses to understand better the relationship between trust and PEOU and PU, on the one hand, and the interaction between PEOU and PU for long-term e-learning adoption:

**Hypotheses 11 (H11).** *Mentality acceptance has a positive moderating effect on the relationship between TOS and PU.*

**Hypotheses 12 (H12).** *Mentality acceptance has a positive moderating effect on the relationship between TOS and PEOU.*

**Hypotheses 13 (H13).** *Mentality acceptance has a positive moderating effect on the relationship between INT and PU.*

**Hypotheses 14 (H14).** *Mentality acceptance has a positive moderating effect on the relationship between INT and PEOU.*

### 3.3. Proposed Research Conceptual Model

To understand the factors affecting the adoption of the e-learning process and to investigate the deployment and acceptance of the system, the proposed research model will help after identifying the main factors that directly affect the adoption of the system and contribute to its long-term sustainability. The proposed research conception model is shown in Figure 1.

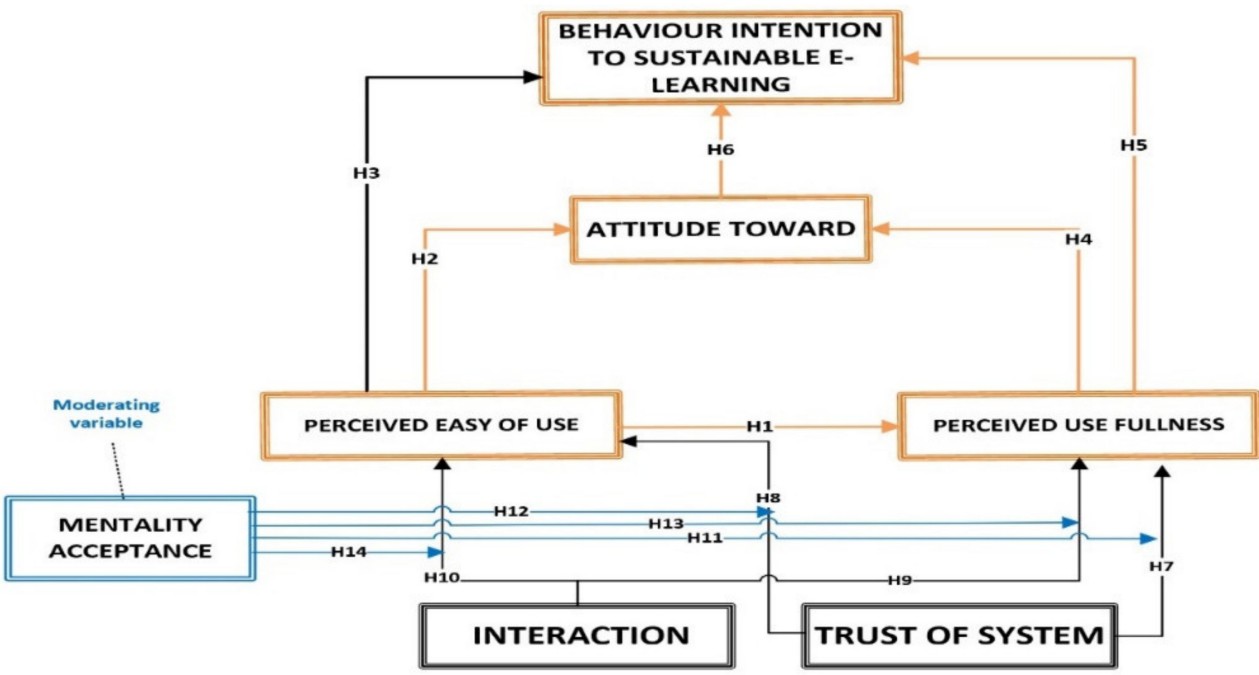

**Figure 1.** Proposed research conceptual model.

## 4. Materials and Methods

Quantitative research methods were used to gauge an e-learning system's potential impact on teachers' and students' attitudes. Because the respondent completes/answers the questionnaire, the interviewer does not need to be present to collect raw data from the interviewee [43]. Questionnaire methodology is the most widely used research method in quantitative studies. The questionnaires were administered to the entire study population. Participants were asked to answer questions that best reflected their views and opinions on various topics, such as the issues at hand [44]. This technique is frequently used to gather primary data due to its simplicity.

In contrast to structured questions, unstructured questions are either open-ended or feature multiple-choice alternatives in a spectrum [45]. In addition, it is simple to administer questionnaires to a large number of individuals. Four hundred students and academic and administrative personnel (e-learning users) were asked to complete online questionnaires. A total of 367 students from educational institutions in northern Iraq participated in the study. Using a 5-point Likert scale, participants were asked to rate each element on a scale of one to five, with one being the most strongly disapproving, three being neutral, and five being the most strongly agreeing. Using SPSS, every piece of data was reviewed before deployment to determine the readiness for e-learning.

## 5. Research Survey Design

### 5.1. Study Questionnaire

Questionnaires are among the tools most commonly used for data collection. The questionnaire used in this study included 35 questions for seven latent constructs that were developed based on previous research [46–48]. It was divided into two sections: the first section consisted of standard questions about the respondent's personal information such as education level, gender, and age, and the second section had questions using a 5-point Likert Scale (i.e., the measurement for each item was performed using a 5-point Likert scale ranging from 1 = strongly disagree to 5 = strongly agree). The second section included five questions for each construct, namely perceived usefulness (PU), perceived ease of use (PEOU), attitude toward technology (ATT), behavioral intention (BI), trust of system (TOS),

mentality acceptance (MA), and interaction (INT). The questionnaire used in this study for data collection is provided in Supplementary Materials.

### 5.2. Designing the Questionnaire

Questionnaires are among the most commonly used structured instruments to obtain numerical information. The research hypotheses were formulated to test the hypotheses. Participants completed this questionnaire by selecting appropriate answers from a list [49]. According to a recent study by the Journal of the American Medical Association, questionnaires are commonly used to acquire information from individuals regarding a wide range of personal and professional issues. A variety of information concerning an individual's thoughts, feelings, attitudes, and behaviors can be obtained from these tests [50]. Consequently, the questions could not be modified after being delivered to participants. Therefore, researchers are unable to ensure honesty. The researchers had no way of checking the responses for correctness [51].

### 5.3. Questionnaire Translation

There are two alternative ways to improve the validity of the questionnaires. In the first step, the questions in Kurdish are translated into English. Another translator, with significant experience, translated the Kurdish version into English. Two Kurdish translators and two English translators checked the translation for accuracy.

### 5.4. Study Sample

As a general rule, samples represent only a small portion of the population or universe at large and are hence called samples. The term "population" conjures up images of humanity. People may not always be counted as a part of the population. Generally, it can also refer to the total number of objects or cases under investigation. Probability sampling is distinguished by the fact that each unit in the population has a non-zero chance of being included in the sample. In other words, everyone who applies has an equal probability of being selected from a pool of applicants. However, randomization was not considered when selecting a sample from the entire population.

Instead, parts included in the study sample were selected using subjective approaches. Students and instructors from all Northern Iraqi governorate schools constituted the sample population. According to the Central Ministry of Education Statistics in Northern Iraq, in 2021, the number of students, administrators, and instructors was 621,342. Some education providers, staff, instructors, and students were required to complete an online questionnaire sent to them. According to Yamane's formula, at the 95% confidence level, the study sample consisted of 400 educational service providers, staff, instructors, and students in Northern Iraq. Out of 400 completed and returned surveys, 33 were discarded as they included unanswered questions. Therefore, in total, 367 valid responses were evaluated.

### 5.5. Respondent Rate and Profile

Table 1 shows that males accounted for 55.9% of the responses, whereas females accounted for 44.1%. The majority of respondents, 54.5 % of the total, were between 15 and 19. The overall response rate in this study was 91.75 %, which is very high in information systems research.

**Table 1.** Demographic summary.

|  | Items | Frequency (*n*) | 100% |
|---|---|---|---|
| Gender | Male | 205 | 55.9 |
|  | Female | 162 | 44.1 |
| Age | 15–19 | 200 | 54.5 |
|  | 20–24 | 6 | 1.6 |
|  | 25–29 | 32 | 8.7 |
|  | 30–34 | 60 | 16.3 |
|  | 35–39 | 33 | 8.1 |
|  | More than 40 | 36 | 9.8 |

## 6. Results

This section is divided into four parts. A concise and precise description of the experimental results, their interpretation, and empirical conclusions can be drawn.

### 6.1. Model Measurement

The reliability test was applied in Table 2 to examine whether or not all of the items utilized in this study were reliable; a Cronbach alpha was calculated for the test reliability scale, consisting of PEOU, ATT, INT, PU, BI, MA, and TOS. The Cronbach's alpha coefficient was evaluated using the guidelines suggested by Alkabaa [52] where 0.9 is excellent, >0.8 is good, >0.7 is acceptable, >0.6 is questionable, >0.5 is poor, and ≤0.5 is unacceptable. Attitudes towards using e-learning have a high level of consistency and are "excellent". The other six combinations in our model were extremely reliable, and all variables were extremely reliable.

**Table 2.** Reliability test of the TAM elements.

| Measurements | Cronbach's Alpha | Number of Items |
|---|---|---|
| Perceived usefulness (PU) | 0.857 | 5 |
| Perceived ease of use (PEOU) | 0.838 | 5 |
| Behavioral intention to use e-learning | 0.891 | 5 |
| Attitudes towards using e-learning | 0.903 | 5 |
| Trust of system | 0.860 | 5 |
| Interaction between user and system | 0.889 | 5 |
| Mentality Acceptance | 0.855 | 5 |
| All | 0.970 | 35 |

### 6.2. Correlation and Relationships

As shown in Table 3, e-learning adoption is strongly linked to each of these characteristics. According to the findings, all constructs have correlations more significant than or equal to 0.5. We found that all the criteria had a beneficial impact on students' attitudes and behaviors surrounding e-learning.

**Table 3.** Correlation matrix between independent variables.

|  | BI | PEOU | PU | ATT | MA | TOS | INT |
|---|---|---|---|---|---|---|---|
| BI | 1 |  |  |  |  |  |  |
| PEOU | 0.776 ** | 1 |  |  |  |  |  |
| PU | 0.762 ** | 0.812 ** | 1 |  |  |  |  |
| ATT | 0.778 ** | 0.812 ** | 0.829 ** | 1 |  |  |  |
| MA | 0.789 ** | 0.793 ** | 0.783 ** | 0.844 ** | 1 |  |  |
| TOS | 0.789 ** | 0.802 ** | 0.780 ** | 0.814 ** | 0.816 ** | 1 |  |
| INT | 0.815 ** | 0.821 ** | 0.829 ** | 0.871 ** | 0.826 ** | 0.824 ** | 1 |
| Mean | 4.23 | 4.28 | 4.22 | 4.22 | 4.25 | 4.23 | 4.27 |
| S.D. | 0.70 | 0.64 | 0.65 | 0.73 | 0.62 | 0.63 | 0.75 |

** Correlation is significant at the 0.01 level (2-tailed).

*6.3. Regression and Hypothesis Test*

To verify the validity and reliability of the measurement model, it was necessary to examine the relationships between the various components. SEM (structural equation modeling) was used to analyze all hypothesized correlations between components, as shown in this section. H1 was tested using multiple regression analysis. The independent variable was perceived usefulness (PU), and the dependent variable was perceived ease of use (PEOU). Table 4 examines the relationship between these two variables. The $R^2$ value was 0.66, indicating that 66% of the variables ewere explained. According to the narrator, this has a significant impact. This research supports H1 and H2 because PU significantly impacts PEOU (=0.81). An F-value of 707.61 > 0.01 showed a positive correlation between the variables. It is expected that an increase in e-learning's perceived ease of use (PEOU) leads to an increase in its perceived utility (PEOU = 0.82 > 0.01).

**Table 4.** The relationship between PU and PEOU.

| Dependent Variable: PEOU | | | | | | |
|---|---|---|---|---|---|---|
|  | **B** | **Std. Error** | **Beta** | **T** | *p* | **Result** |
| Constant | 0.69 | 0.13 |  | 5.11 | 0.001 |  |
| PEOU | 0.82 | 0.031 | 0.81 | 26.6 | <0.001 | Supported |
| Model F | 707.61 |  |  |  |  |  |
| $R^2$ | 0.66 |  |  |  |  |  |

*p* < 0.05.

A positive correlation between PEOU (=0.93) and students' attitudes toward using e-learning (ATT) at 0.5 is shown in Table 5 as support for H2.

**Table 5.** The relationship between PEOU and ATTITUDE.

| Dependent Variable: ATTITUDE | | | | | | |
|---|---|---|---|---|---|---|
|  | **B** | **Std. Error** | **Beta** | **T** | *p* | **Result** |
| Constant | 0.23 | 0.15 |  | 1.51 | 0.132 |  |
| PEOU | 0.93 | 0.04 | 0.81 | 26.54 | <0.001 | Supported |
| Model F | 704.36 |  |  |  |  |  |
| $R^2$ | 0.66 |  |  |  |  |  |

*p* < 0.05.

As the single regression analysis demonstrated, the correlation coefficient (F = 704.36 >0.01) suggests a positive relationship between BI's variables for perceived ease of use. PEOU = 0.93 (>0.01) indicates that improving the PEOU of e-learning will lead to an increasingly positive attitude towards e-learning. PEOU = 0.93 (0.93 > 0.01) suggests that students' attitudes toward online learning will improve as the perceived ease of use increases. $R^2$ = 0.66 indicates that 66% of the variables were explained.

PEOU and behavioral intention toward adopting e-learning, as proposed by the TAM, explained a large amount of variance in attitude ($R^2$ = 0.60) (Table 6). PEOU significantly influenced BI (=0.78), indicating that the third research hypothesis was supported. The value F = 553.83 > 0.01 suggests that the variables have a positive connection, with the value B showing PEOU. PEOU = 0.85 (0.85 > 0.01) indicated that increasing the perceived ease of use of e-learning enhances behavioral intention to use it.

**Table 6.** The relationship between PEOU and BI.

| | Dependent Variable: BI | | | | | |
|---|---|---|---|---|---|---|
| | **B** | **Std. Error** | **Beta** | **T** | ***p*** | **Decision** |
| Constant | 0.58 | 0.16 | | 3.68 | 0.001 | |
| PEOU | 0.85 | 0.04 | 0.78 | 23.53 | <0.001 | Supported |
| Model F | 553.83 | | | | | |
| $R^2$ | 0.729 | | | | | |

*p* < 0.05.

The fourth hypothesis examined the effect of perceived usefulness on attitudes. As shown in Table 7, PU had a direct positive influence on attitude (=0.83, *p* < 0.05), which is similar to prior TAM investigations [6]. Therefore, H4 was recommended. H5 and H6 examine the relationships between PU, ATT, and BI. As shown in Tables 8 and 9, PU had a favorable direct effect on behavioral intention (=0.76, *p* < 0.05) and attitudes toward e-learning (=0.78, *p* < 0.05). Thus, H5 and H6 were accepted.

**Table 7.** The connection between PU and ATTITUDE.

| | Dependent Variable: ATTITUDE | | | | | |
|---|---|---|---|---|---|---|
| | **B** | **Std. Error** | **Beta** | **T** | ***p*** | **Result** |
| Constant | 0.27 | 0.14 | | 1.89 | 0.059 | |
| PU | 0.94 | 0.03 | 0.83 | 28.29 | 0.000 | Supported |
| Model F | 450.963 | | | | | |
| $R^2$ | 0.800 | | | | | |

*p* < 0.05.

**Table 8.** The connection between PU and BI.

| | Dependent Variable: BI | | | | | |
|---|---|---|---|---|---|---|
| | **B** | **Std. Error** | **Beta** | **T** | ***p*** | **Result** |
| Constant | 0.75 | 0.16 | | 4.81 | <0.001 | |
| PU | 0.82 | 0.04 | 0.76 | 22.48 | <0.001 | Supported |
| Model F | 505.33 | | | | | |
| $R^2$ | 0.58 | | | | | |

*p* < 0.05.

**Table 9.** The relationship between Attitude and BI.

| | B | Std. Error | Beta | T | *p* | Result |
|---|---|---|---|---|---|---|
| | | | **Dependent Variable: BI** | | | |
| Constant | 1.09 | 0.014 | | 8.08 | <0.001 | |
| ATTITUDE | 0.74 | 0.03 | 0.78 | 23.64 | <0.001 | Supported |
| Model F | 558.68 | | | | | |
| $R^2$ | 0.60 | | | | | |

*p* < 0.05.

As shown in Table 10, the 7th hypothesis test (H7) used trust in the system (TOS) as an independent variable and perceived usefulness (PU) as the dependent variable. The $R^2$ was 0.61; 61% of the variables were explained. TOS strongly influenced PU (=0.78), supporting H18. F = 565.96 > 0.01, B = 0.80, t (365) = 23.79, *p* < 0.001. Increasing the TOS by one-unit increased PU by 0.80.

**Table 10.** The connection between TOS and PU.

| | B | Std. Error | Beta | T | *p* | Result |
|---|---|---|---|---|---|---|
| | | | **Dependent Variable: PU** | | | |
| Constant | 0.83 | 0.14 | | 5.76 | <0.001 | |
| TOS | 0.80 | 0.03 | 0.78 | 23.79 | <0.001 | Supported |
| Model F | 655.85 | | | | | |
| $R^2$ | 0.64 | | | | | |

*p* < 0.05.

In Table 11, it can be seen that test H8 was dependent on system trust (TOS) and perceived ease of use (PEOU) for the independent and dependent variables, respectively. R2 explained 64% of the variables. TOS had a considerable impact on PEOU (as defined above) and supports H19. As evidence, F = 655.85 < 0.01, B = 0.81, and t (365) = 25.61, all with *p* < 0.001. The TOS must be increased by one unit to achieve a 0.81-unit increase in PEOU.

**Table 11.** The connection between TOS and PEOU.

| | B | Std. Error | Beta | T | *p* | Result |
|---|---|---|---|---|---|---|
| | | | **Dependent Variable: PEOU** | | | |
| Constant | 0.85 | 0.14 | | 6.29 | <0.001 | |
| TOS | 0.81 | 0.03 | 0.80 | 25.61 | <0.001 | Supported |
| Model F | 655.85 | | | | | |
| $R^2$ | 0.64 | | | | | |

*p* < 0.05.

Perceived usefulness (PU) of e-learning was studied with INT as an independent variable. The $R^2$ value was 0.69, meaning that 69% of the variables were explained. This has a significant effect (Anderson and Gerbing, 1988). INT strongly influenced PU (0.83), thus supporting H9 (Table 12). F = 804.13 > 0.01, B = 0.72, t (365) = 28.36, and *p* < 0.001. Increasing INT by one unit increased the PU by 0.72 units.

**Table 12.** The connection between INT and PU.

| | Dependent Variable: PU | | | | | |
|---|---|---|---|---|---|---|
| | **B** | **Std. Error** | **Beta** | **T** | **_p_** | **Result** |
| Constant | 1.13 | 0.11 | | 10.26 | <0.001 | |
| INT | 0.72 | 0.03 | 0.83 | 28.36 | <0.001 | Supported |
| Model F | 804.13 | | | | | |
| $R^2$ | 0.69 | | | | | |

$p < 0.05$.

Perceived ease of use toward e-learning (PEOU) was the dependent variable in the 21st hypothesis test's (H10) linear regression analysis (Table 13). The R2 value was 0.67, indicating that 67% of the variables were explained. This has a significant effect. INT strongly influences PEOU (=0.82), supporting H11. F = 756.09 > 0.01, B = 0.70, t (365) = 27.50, and $p < 0.001$. This means that a one-unit increase in INT increases PEOU by 0.70 units.

**Table 13.** The connection between INT and PEOU.

| | Dependent Variable: PEOU | | | | | |
|---|---|---|---|---|---|---|
| | **B** | **Std. Error** | **Beta** | **T** | **_p_** | **Result** |
| Constant | 1.28 | 0.11 | | 11.47 | <0.001 | |
| INT | 0.70 | 0.03 | 0.82 | 27.50 | <0.001 | Supported |
| Model F | 756.09 | | | | | |
| $R^2$ | 0.67 | | | | | |

$p < 0.05$.

*6.4. Moderation Analysis*

As a support requirement for moderation, the following must be met [53]. First, the causative variable must significantly predict the predictor variable (step 1). Second, the interaction model must explain substantially more of the variance in the predictor variables than the non-interaction model (step 2) Both these factors must be met for moderation to be recommended. These regressions were analyzed based on an alpha of 0.05.

TOS significantly predicted PU (B = 0.80, t (365) = 23.79, $p < 0.001$. Therefore, the first condition was met, and the second condition was checked. The partial F-test, F (1363) = 68.23, $p < 0.001$, indicated that the interaction model explained significantly more variance than the non-interaction model, based on an alpha of 0.05. Therefore, the second condition is satisfied. Since TOS significantly predicted PU in the simple effects model (condition 1) and the interaction model explained significantly more variance in PU than the non-interaction model (condition 2), moderation is supported. The results of the simple, non-interaction, and interaction models are presented in Table 14. Table 15 presents a comparison of the non-interaction and interaction models. Based on an alpha of 0.05, B = −0.25, t (363) = −8.26, and $p < 0.001$, MA significantly moderated the effect of TOS on PU, thus supporting H11. This indicates that, on average, a one-unit increase in MA causes a 0.25 decrease in the slope of PU on TOS. MA was dichotomized into high and low categories to visualize the moderation analysis using a median split. The high category indicates that all MA observations are above the median, whereas the low category indicates that all MA observations are below the median. Additionally, for the moderating role of MA in the relationship between TOS and PEOU to be supported, two conditions must be met, and the regressions must be examined based on an alpha of 0.05. TOS significantly predicted PEOU (B = 0.81, t (365) = 25.61, and $p < 0.001$). Therefore, the first condition was met, and the second condition was checked through a partial F-test, F (1363) = 47.84, $p < 0.001$. The result indicated that the interaction model explained significantly more variance than the non-interaction model based on an alpha of 0.05; thus, moderation is supported. The results

of the simple, non-interaction and interaction models are presented in Table 16. Table 17 presents a comparison of the non-interaction and interaction models. MA significantly moderated the effect of TOS on PEOU based on an alpha of 0.05, B = −0.20, t (363) = −6.92, and *p* < 0.001, and supported H12.

**Table 14.** Moderation analysis with PU predicted by TOS moderated by MA.

| Predictor | B | SE | β | t | *p* |
|---|---|---|---|---|---|
| Step 1: simple effects model | | | | | |
| Intercept | 0.83 | 0.14 | | 5.76 | <0.001 |
| TOS | 0.80 | 0.03 | 0.78 | 23.79 | <0.001 |
| Step 2: non-interaction model | | | | | |
| (Intercept) | 0.44 | 0.14 | | 3.15 | 0.002 |
| TOS | 0.43 | 0.05 | 0.42 | 8.11 | <0.001 |
| MA | 0.46 | 0.05 | 0.44 | 8.51 | <0.001 |
| Step 3: interaction model | | | | | |
| Intercept | 4.31 | 0.02 | | 210.59 | <0.001 |
| TOS | 0.23 | 0.05 | 0.23 | 4.24 | <0.001 |
| MA | 0.27 | 0.05 | 0.26 | 4.95 | <0.001 |
| TOS:MA | −0.25 | 0.03 | −0.42 | −8.26 | <0.001 |

**Table 15.** Linear model comparison between the non-interaction and interaction model.

| Model | $R^2$ | F | df | *p* |
|---|---|---|---|---|
| Non-interaction | 0.67 | | | |
| Interaction | 0.72 | 68.23 | 1 | <0.001 |

**Table 16.** Moderation analysis with PEOU predicted by TOS moderated by MA.

| Predictor | B | SE | β | t | *p* |
|---|---|---|---|---|---|
| Step 1: simple effects model | | | | | |
| (Intercept) | 0.85 | 0.14 | | 6.29 | <0.001 |
| TOS | 0.81 | 0.03 | 0.80 | 25.61 | <0.001 |
| Step 2: non-interaction model | | | | | |
| (Intercept) | 0.49 | 0.13 | | 3.72 | <0.001 |
| TOS | 0.47 | 0.05 | 0.46 | 9.32 | <0.001 |
| MA | 0.43 | 0.05 | 0.42 | 8.39 | <0.001 |
| Step 3: interaction model | | | | | |
| (Intercept) | 4.36 | 0.02 | | 220.57 | <0.001 |
| TOS | 0.31 | 0.05 | 0.30 | 5.80 | <0.001 |
| MA | 0.27 | 0.05 | 0.27 | 5.20 | <0.001 |
| TOS:MA | −0.20 | 0.03 | −0.35 | −6.92 | <0.001 |

**Table 17.** Linear model comparison between the non-interaction and interaction model.

| Model | R$^2$ | F | df | *p* |
|---|---|---|---|---|
| Non-interaction | 0.70 | | | |
| Interaction | 0.74 | 47.84 | 1 | <0.001 |

Moreover, INT significantly predicted PU (B = 0.72, t (365) = 28.36, and *p* < 0.001). The INT significantly predicted PU. The results of the simple, non-interaction, and interaction models are presented in Table 18. Table 19 presents a comparison of the non-interaction and interaction models. MA significantly moderated the effect of INT on PU based on an alpha of 0.05, B = −0.18, t (363) = −6.18, and *p* < 0.001, supporting H13. Finally, to determine the moderating role of MA in the relationship between INA and PEOU, regressions are examined based on an alpha of 0.05. PEOU was significantly predicted by INT (B = 0.70, t (365) = 27.50, and *p* < 0.001). The results of the simple, non-interactive, and interactive models are presented in Table 20. Table 21 presents a comparison of the non-interaction and interaction models. Based on an alpha of 0.05, B = −0.16, t (363) = −5.52, and *p* < 0.001, MA significantly moderated the effect of INT on PEOU, thus supporting H14.

**Table 18.** Moderation analysis with PU predicted by INT and moderated by MA.

| Predictor | B | SE | β | t | *p* |
|---|---|---|---|---|---|
| Step 1: simple effects model | | | | | |
| (Intercept) | 1.13 | 0.11 | | 10.26 | <0.001 |
| INT | 0.72 | 0.03 | 0.83 | 28.36 | <0.001 |
| Step 2: non-interaction model | | | | | |
| (Intercept) | 0.71 | 0.12 | | 5.72 | <0.001 |
| INT | 0.50 | 0.04 | 0.57 | 11.62 | <0.001 |
| MA | 0.32 | 0.05 | 0.31 | 6.28 | <0.001 |
| Step 3: interaction model | | | | | |
| (Intercept) | 4.29 | 0.02 | | 207.90 | <0.001 |
| INT | 0.26 | 0.06 | 0.30 | 4.62 | <0.001 |
| MA | 0.22 | 0.05 | 0.22 | 4.35 | <0.001 |
| INT:MA | −0.18 | 0.03 | −0.39 | −6.18 | <0.001 |

**Table 19.** Linear model comparison between the non-interaction and interaction models.

| Model | R$^2$ | F | df | *p* |
|---|---|---|---|---|
| Non-interaction | 0.72 | | | |
| Interaction | 0.75 | 38.22 | 1 | <0.001 |

**Table 20.** Moderation analysis table with PEOU predicted by INT and moderated by MA.

| Predictor | B | SE | β | t | *p* |
|---|---|---|---|---|---|
| Step 1: simple effects model | | | | | |
| (Intercept) | 1.28 | 0.11 | | 11.47 | <0.001 |
| INT | 0.70 | 0.03 | 0.82 | 27.50 | <0.001 |
| Step 2: non-interaction model | | | | | |
| (Intercept) | 0.79 | 0.12 | | 6.42 | <0.001 |

**Table 20.** *Cont.*

| Predictor | B | SE | β | t | p |
|---|---|---|---|---|---|
| INT | 0.45 | 0.04 | 0.52 | 10.55 | <0.001 |
| MA | 0.37 | 0.05 | 0.36 | 7.28 | <0.001 |
| Step 3: interaction model | | | | | |
| (Intercept) | 4.35 | 0.02 | | 210.91 | <0.001 |
| INT | 0.24 | 0.06 | 0.27 | 4.18 | <0.001 |
| MA | 0.28 | 0.05 | 0.28 | 5.49 | <0.001 |
| INT:MA | −0.16 | 0.03 | −0.36 | −5.52 | <0.001 |

**Table 21.** Linear model comparison between the non-interaction and interaction model.

| Model | $R^2$ | F | df | p |
|---|---|---|---|---|
| Non-interaction | 0.72 | | | |
| Interaction | 0.74 | 30.46 | 1 | <0.001 |

## 7. Discussion

TAM explains how students' cultural reactions follow a specific pattern. According to Hypotheses H1–H3, PEOU positively impacts the usefulness of e-learning perceptions, attitudes toward e-learning, and behavioral intention to use the system. PEOU is critical, particularly during the initial stages of implementing new technologies. Previous studies have found that if a person is already familiar with new technology, they are more likely to accept it easily [21]. Hypotheses H4 and H5 predicted a positive attitude toward e-learning and positive behavioral intention to employ e-learning. These two hypotheses were proven true: individuals are more likely to adopt new technology when they perceive it to provide benefits that they can appreciate. TAM research relies heavily on utility perceptions, influencing attitudes and motivations towards using new technology [54]. Attitude towards e-learning positively influences behavioral intention to employ e-learning (H6). The students' opinions and attitudes directly impacted whether they wanted to use e-learning technology. As Davis, Bagozzi, and Warshaw [6] state, "attitude influences the intention to employ new technology and is critical for TAM research" [55]. The system trust factor predicted perceived e-learning utility (=0.780, $p < 0.001$), thus supporting H7.

H8 was supported by the system trust factor (=0.802, $p < 0.001$), which predicted the perceived ease of use of e-learning for adoption. Similarly, the perceived utility of e-learning in terms of adoption was strongly predicted by the system interaction factor (=0.829, $p < 0.001$), validating the study's H9. TAM studies have shown that a lack of trust in the system has a significant impact on participants' intentions to use new technology, affecting both their perceptions of the usefulness and ease of use of the technology [31,32,56]. H10 was supported by the interaction factor (=0.821, $p < 0.001$), which predicted the perceived ease of use of e-learning for implementation. In TAM research, system interaction has been shown to negatively impact the perceived usefulness and ease of use of new technology in TAM research [33,57]. Finally, the acceptance component of mentality had a moderating influence. In H11, the acceptance of mentality moderated the association between system trust and the utility of e-perceived learning for adoption. In H11, trust in a system's perceived ease of use for e-learning implementation is positively influenced by mental acceptance, thereby supporting H12.

New technology is more likely to be adopted by users who believe it will be more beneficial than the current options. According to H13, there is a favorable association between system interaction and the perceived usefulness of e-learning for adoption. H14 is supported by the mentality acceptance factor, which modifies the relationship between

system interaction and perceived ease of use in e-learning implementation. People perceive the advantages of new technologies and are more likely to use them.

## 8. Conclusions

The fundamental purpose of this study was to identify and examine the factors affecting the adoption and sustainability of e-learning systems in this digital era. Through the various studies discussed in the literature review section, it is understandable that no proper study was conducted regarding the implication of ICT usage in places such as Northern Iraq. This study considered this research gap and, by using the TAM approach, surveyed 367 instructors and students from MOE schools in Northern Iraq's Ministry of Education to determine their attitudes towards education. MOE officials, staff, instructors, and students provided data in an online survey. The SPSS statistical package was used to analyze the data collected from the study participants who were chosen randomly.

TAM research has shown that system mentality acceptance has a significant impact on perceptions of usefulness and ease of use, which affects willingness to adopt new technologies [39,42]. In the early phases of implementing new technology, PEOU is essential. When a person is already familiar with technology, they are more likely to accept it [58]. The findings of other researchers were consistent with this finding [26]. According to the results, better adoption rates can be attributed to students' and teachers' confidence in the system [30].

According to the findings of this study, attitudes and behavior are positively influenced by the intention to use e-learning services because of their perceived simplicity of use and usefulness. Furthermore, the results showed that PU and PEOU were positively impacted by trust in the system and interaction factors. A statistically significant difference between age and educational level was found in this study, with the exclusion of gender and the intention to use e-learning.

## 9. Practical and Theoretical Implications

In recent years, the education model of the world has suffered a lot due to the pandemic. As many countries have not yet adopted e-learning, students and teachers are facing a lot of difficulties during the teaching–learning process. This study presents empirical evidence for novel contributions to the practical and theoretical ramifications. In terms of theoretical implications, the TAM model was effectively improved and implemented considering e-learning in Northern Iraq. The TAM model was reformed to include the most common characteristics such as mentality acceptance, interaction, and trust of the system. The empirical results prove that the developed conceptual model is operative in understanding behavioral intention in adopting an e-learning environment. According to our knowledge, this is the first study to examine the factors that are influencing e-learning adoption in Northern Iraq considering the challenges faced during the pandemic. The government can infer the knowledge about the imperative of E-learning and can develop a framework for the adoption of ICT-based education models in the educational institutions. The results of this study will also help private and public educational institutions to understand the importance of imparting the ICT-based education model in their teaching–learning process, which can support them in achieving a sustainable educational model in underdeveloped and developing countries. From the practical implications perspective, to maintain a positive intention to use e-learning systems, developers should focus on developing applications that can work on various types of communication devices with internet accessibility, improve the system performance and create user-friendly interfaces. In addition, from the perspective of educational adopters, students' behavioral intentions towards e-learning apps can be measured by identifying the factors and impediments for the sustainable adoption of e-learning using the proposed model in this study.

## 10. Limitations and Future Studies

This study provides new theoretical and practical contributions that are supported empirically. In addition, the moderating effect of the relationship between trust in the system and the perceptions of ease of use and utility was considered. This study's limitation is that it did not consider how the relationships among system interaction, perceived usability, and ease of use can operate as moderators. Future studies can consider factors and methods for implementing ICT-based education in other developing and underdeveloped countries. The proposed model can be improved by additional variables. Gender differences can also be evaluated in future studies.

**Supplementary Materials:** The following are available online at https://www.mdpi.com/article/10.3390/su14063690/s1.

**Author Contributions:** Data curation, S.S.S. and M.N.; Investigation, M.A.; Methodology, S.S.S.; Project administration, S.S.S.; Resources, S.S.S.; Supervision, M.N. and M.A.; Writing—original draft, S.S.S.; Writing—review & editing, S.S.S. and M.A. All authors have read and agreed to the published version of the manuscript.

**Funding:** This research received no external funding.

**Institutional Review Board Statement:** The study was conducted in accordance with the Declaration of Helsinki, and approved by the Institutional Review Board (or Ethics Committee) of Cyprus International University.

**Informed Consent Statement:** Not applicable.

**Conflicts of Interest:** The authors declare no conflict of interest.

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
