# Peer review of "Sustainable Adoption of E-Learning from the TAM Perspective"

_sustainability, doi:10.3390/su14063690_

Round 1

Reviewer 1 Report

This research is relevant due to the active use of modern technologies in the educational process. The results of the study contribute to the methodology of teaching using modern technologies. The structure of the study is clear and clear. Despite the large number of citations that slightly reduce the originality of the study, the study itself is interesting and useful for teachers.

Author Response

Dear Reviewer,

Thank you for your valuable time and effort in reviewing the article. Your valuable comments helped us in improving the quality of the article. The point by point responses for the queries is given as a separate file.

Thanks again 

Reviewer 2 Report

Overall, I really appreciate learning a little bit more about adopting e-learning from the TAM perspective in your context. However, I feel that the manuscript could be improved to showcase better the work you have accomplished. Specifically, I think a clearer and more concise style could strengthen the writing style. Many of your sentences incorporate informal phrases and redundant words (I have highlighted some below). In other words, a more robust academic writing style could be applied in the article.  

Introduction

Abbreviation: Spell out “ICT” the first time it is used (line 33)

Informal phrases: to put it another way (line 34)

Line 40 – the sentence is awkwardly phrased and lack logic

Line 41 – the word “worldwide” is redundant

Line 54 – full stop after (TAM). Not before

The introduction should be shortened to 1-2 paragraphs. The left-over materials can be included in the revised literature review.

Research Context

Line 94 -95 what does “still in the process of being developed” actually mean?

Line 95 – what do you mean by “real world”

Line 101. Should there be citation as evidence for the claim

Research Question: Don’t hide it in a paragraph—separate line.

Also, you say the main research question. What are the other research questions?

Materials and Method

4.1 Study questionnaire

This section doesn’t talk about the questionnaire in this study. It is suggested the authors include a rationale for using their questionnaire, how it was developed/adopted, etc. Instead, they have sprinkled some of it in each very confusing section.

Each section on Materials and Method need to be rewritten with a clear focus on the actual purpose and methods.

Quite confusing why the authors are discussing

Result

Clear

Discussion

The authors have done a good to interpreting and describing the significance of their findings, but more in-depth discussion should be included. Specifically, what was already known about the research problem being investigated and any new understandings or insights were explained. In other words, more details should be incorporated with explicit references to the literature. This would help to highlight the importance of your study and how it can contribute to understanding the research problem within the field of study.

Conclusion

The conclusion has not identified how a gap in the literature has been addressed. The authors need to consider/describe how their study has filled a void. Though the authors have summarized the main findings, no citations have included the literature review. They did not demonstrate the importance of their ideas/findings and elaborated on the impact and significance of their results.

Implications to theory and practice

This section needs to be “fleshed out” to provide concrete implications for both theory and practice. Currently, the idea(s) are there; however, they are too vague.

Limitations and Future Studies

This should be included as a separate section  

Author Response

(The authors gave the same response as above.)

Reviewer 3 Report

The authors present an interesting article with a good approach and development of the investigation. However, in the results section a large number of tables (more than 20) are observed. The article would be more enriching if some of the data and information presented in tables were collected in the text itself.

Author Response

(The authors gave the same response as above.)

Reviewer 4 Report

I am pleased to have the opportunity to review this research paper. Although the topic of this research study is interesting and fits within the journal scope, I think authors should apply the comments indicated below to increase the quality of research justification, contributions and findings.

The similarity index is a little bit high maybe you can reduce it. I attached the similarity report to help you to improve your paper.

The table must be reforming according to the journal.

The Materials and Methods can be improved with more detailed information to help the reader to understand it better.

I suggest to writ a sort comment after each table to understand what the table illustrate and to have a better logical flow.  

Reference style must be improved according to the journal, and you maybe can add some references like:

https://doi.org/10.3390/su13042052

https://doi.org/10.3390/ijerph18147533

https://doi.org/10.3390/app11052365

Please consider this structure for manuscript final part.
-Discussion
-Conclusion
-Managerial Implication
-Practical/Social Implications

Good luck!

Author Response

(The authors gave the same response as above.)

Round 2

Reviewer 2 Report

Introduction

What’s the purpose of the first sentence in the introduction? How does it fit with the scope of the paper?

Authors keep referring to mobile commerce – is the paper about mobile commerce?

In-text references are not in the correct order.

  1. Research Design

4.1 Study Questionnaire

Majority of information in this paragraph is irrelevant. Provide suitable justification/rationale for the questionnaire items.

Missing words on line 305

This entire section needs to be rewritten.

---------

Include the questionnaire and interview questions in the appendix

Practical and Theoretical Implications

This needs to be further teased out and expanded to actually illustrate the findings from the study. Currently, both the practical and theory (not mentioned) is too vague.

Author Response

Dear Reviewer,

We thank you for your time and effort in reviewing the article. Your points really help us in improving the quality of the article. We hope that the revised article is addressing all your queries satisfactorily. The response for each query is given as a separate.

Thanks again

Reviewer 4 Report

Thank you very much for your work.

Just a remark in line 305 "Questionnaire used in this study is adopted from …. . Questionnaire consists of seven sections, and each section has five questions." I think you must fill with references or something the .....

Congratulations for the material and I wish you good luck!

Author Response

Dear Reviewer,

We thank you for your time and effort in reviewing the article. Your points really help us in improving the quality of the article. We hope that the revised article is addressing all your queries satisfactorily. The response for each query is given as a separate file.

Thanks again
